# Catalyzing Transformational Partnerships for the SDGs: Effectiveness and Impact of the Multi-Stakeholder Initiative *El día después*

**Jaime Moreno-Serna** [1], **Wendy M. Purcell** [2], **Teresa Sánchez-Chaparro** [1,3,*], **Miguel Soberón** [1], **Julio Lumbreras** [1,4,5] **and Carlos Mataix** [1,3]

1   Centro de Innovación en Tecnología para el Desarrollo Humano, Universidad Politécnica de Madrid (itdUPM), 28040 Madrid, Spain; jaime.moreno@upm.es (J.M.-S.); miguel.soberon@upm.es (M.S.); julio.lumbreras@upm.es (J.L.); carlos.mataix@upm.es (C.M.)

2   Harvard T.H. Chan School of Public Health, Harvard University, Boston, MA 02215, USA; wpurcell@hsph.harvard.edu

3   Department of Organizational Engineering, Business Administration and Statistics, Escuela Técnica Superior de Ingenieros Industriales, Universidad Politécnica de Madrid, 28040 Madrid, Spain

4   Harvard J. F. Kennedy School of Government, Harvard University, Cambridge, MA 02138, USA

5   Department of Chemical Engineering and Environment, Industrial Engineering School, Universidad Politécnica de Madrid, 28040 Madrid, Spain

*   Correspondence: teresa.sanchez@upm.es; Tel.: +34-910-671-397

**Abstract:** Partnerships are essential to delivering the transformational change demanded by the Sustainable Development Goals (SDG) and essential to achieving Agenda 2030. It is therefore necessary to strengthen the partnering capacity of different types of organizations so they can collaborate in multi-stakeholder partnerships. However, partnership working can be costly in terms of time and other resources and is complex. Given the urgency and importance of sustainable development, illustrated by the recent pandemic and social unrest around inequity, we focused on the creation of a partnership that became effective quickly and was able to deliver societal impact at scale. Using a case study approach, the transformational potential and the early stages of "El día después" (in English, "The day after") were analyzed as it represents a multi-stakeholder partnership forged to frame an SDG-oriented collaborative response to the COVID-19 crisis in Spain. *El día después* is defined as a partnership incubator, a space where public administrations interact under conditions of equivalence with all the other stakeholders, where private companies can link their innovation processes to other SDG-committed actors and social needs and where the academic sector can participate in a sustained dialogue oriented to the action. Our findings reveal that in order to catalyze the co-creation process and achieve systemic change through a set of connected multi-stakeholder initiatives, a very flexible collaborative arrangement is required, with all partners acting as facilitators. In this way, a solid interdisciplinary team is created, united around a shared vision, with trust-based relationships and a common identity fueling impact-oriented projects targeted to advance the SDGs.

**Keywords:** multi-stakeholder partnerships; transformation; effectiveness; impact; SDGs; COVID-19

## 1. Introduction

The adoption in 2015 of the 2030 Agenda for Sustainable Development and its 17 Sustainable Development Goals (SDGs) revealed the need for strengthen global partnerships. 'Effective public, public-private and civil society partnerships' highlighted in Target 17.1 may result the institutional and organizational structures needed to foster the systemic and transformative approaches required

to deliver against the SDG Agenda [1–4]. "These transformations seek to exploit synergies between Goals to achieve multiple SDGs by organizing implementation around SDG interventions that generate significant co-benefits" [4] (p. 2). Stronger governance structures may emerge as a result of the exploitation of these synergies [4–6].

Given the urgency to advance sustainable development, highlighted by the recent pandemic and social unrest around inequity, we need to be more deliberate in creating multi-stakeholder partnerships and pay more attention to the ingredients that promote effectiveness and impact through partnership working. Without this, we shall continue to rely on serendipity and opportunism to bring partners together [7]. An essential question to address is "Who starts it?" Partnerships that have a transformative ambition cannot rely on bottom-up approaches alone [8,9], which can present problems such as short-time horizons, insufficient coordination mechanisms and misaligned incentives [4]. To achieve systemic impact, it is necessary to gain wide agreement on the transition pathway or roadmap and the portfolio of partnerships needed in order to achieve it [4]. Facilitators may bring partners together, help with the incentives assessment or assist in any conflict resolution [5,10]; however, much enquiry is needed around how to develop a collaborative roadmap and the nature of partnerships needed to create a suitable portfolio [4]. It takes energy to both initiate, develop and sustain partnership working, and the return on this investment of time, personnel and other resources needs to be worth the effort [11–13]. Usually, partners have clear incentives in terms of efficiency, innovation or reputation [14–17]. However, partnership processes can be resource and time demanding and more practical evidence is needed on how to make them more truly effective and impactful [4].

Here, we explore the deliberate creation of a large multi-stakeholder partnership from a lifecycle perspective, paying close attention to how the formative stages of relationship building were accelerated, identifying the key ingredients required and how the partnership moved beyond incrementalism to deliver transformative change. We focused on drawing out how the partnership became a solid initiative, with value beyond that of the sum of the partners. The case of "El día después" (EDD; in English, "The day after"), forged to frame an SDG-oriented response to the COVID-19 crisis, represents a unique partnership devoted to building the capacity of relevant stakeholders to tackle the pandemic crisis. EDD was used as a vehicle in which to examine the establishment of a partnership at speed, charged with a higher degree of transformation and focused on delivering more impact.

The article is organized as follows: Section 2 provides a theoretical overview of partnerships, their lifecycle and how they may transition to transformational status. Section 3 presents the research approach, based on a case study methodology. To assist better understanding of the subsequent analysis, a summary of the results and organizational model of EDD is provided in Section 4. In Section 5, a detailed analysis of the initial phases of the partnership lifecycle, and an assessment of its transformational character is provided. Key conclusions and lessons are presented in Section 6, including some recommendations about creating partnering capacity around relevant stakeholders to accelerate the transformations needed to achieve SDGs.

## 2. Theoretical Overview

Collaboration among different stakeholders in society seeks to assemble diverse and potentially complementary assets, in the form of competencies, skills and resources, around a shared purpose that guides their attention—in this case, transformation related to delivery of the 2030 Agenda [2,3]. While the study of the processes necessary for partnerships to generate systemic change has been the subject of academic enquiry, it is necessary to explore specific cases to further theoretical analysis of their lifecycle [4–6,18]. There is general agreement that three elements are critical to a partnership seeking to achieve systemic change:

- The formation stage of the partnership [11,12,14,19] represents a period of intense investment by the partners to define the value-add of the collaboration, to develop trust among them and to set the goals and systems for working together [20,21].

- Articulating the aspiration of the partners for transformational change. The collaborative value [15] created at the initial stages of a collaboration usually rests upon philanthropic or transactional approaches, with several critical factors identified for a partnership to evolve to a transformational stage [15].
- The need for orchestration or a facilitation function, with governance processes that assist partners and wider stakeholders to manage and respond to the challenges of collaboration [4,5,10].

A number of researchers have examined partnerships and collaborative arrangements using a chronological approach [17,22]; this can help us understand the processes that enable transformational outcomes [23] across key phases that partnerships normally go through, although, in most cases, progression is non-linear and phases overlap [24]. They proposed a cyclical process that begins with scoping, where the challenge to be tackled is identified and the partners selected. This is followed sequentially by setting the objectives, roles and governance structures. The third phase relates to implementation and is oriented towards action, when partners' engagement and appropriate collaborative management are crucial. After this, the partnership becomes consolidated and moves to either complete its project and/or transitions to tackle new work together. Key attributes of the initial stages [24] are summarized in Table 1.

**Table 1.** Initial phases of partnership development [24].

| Stage | Key Attributes |
|---|---|
| Scoping | Purpose and orientation Composition Articulation |
| Initiating | Agreements and decision-making Partners expansion |
| Implementing | Launching Operation Scaling up strategies |

Within this lifecycle framework, the importance of the scoping and initiating stages has been widely acknowledged [20,21]. In these early phases, the partnership may be less visible, with resources being consumed and value yet to be created and/or made visible [11–13]. The opportunity to pay attention to "value creation dynamics" [15] may help to attenuate or indeed avoid downstream challenges to effective collaboration [13,14].

A framework that can help us understand the purpose of partnerships that reach across business and society is that of shared value [25], developed to illustrate the policies and operating practices that enhance the competitiveness of a company while simultaneously advancing the economic and social conditions in the communities in which it operates. Some limitations were identified with this concept, drawing attention to the tensions between economic and social objectives and the lack of an overt link to social innovation [26,27]. Proposed set of tools which conceptualize key elements and processes in fostering shared value through cross-sector partnerships, defined the collaborative value creation (CVC) framework [15]. The CVC framework identifies four stages of collaboration that progress one to the next as partners reinforce the generation of meaningful shared value (see Figure 1). The CVC framework brings two fundamental elements to the conceptualization of partnerships: the transformational aspiration among the partners and the evolutionary nature of the value generated. However, its focus on companies and non-profits poses some limitations in terms of the diversity of actors comprising a partnership; the emerging facilitating or orchestrating role in a partnership was also not considered explicitly.

| NATURE OF RELATIONSHIP | StageI: PHILANTROPIC | StageII: TRANSACTIONAL | StageIII: INTEGRATIVE | StageIV: TRANSFORMATIONAL |
|---|---|---|---|---|
| Level of engagement | Low | | | High |
| Importance to mission | Peripheral | | | Central |
| Magnitude of resources | Small | | | Big |
| Type of resources | Money | | | Core competences |
| Scope of activities | Narrow | | | Broad |
| Interaction level | Infrequent | | | Intensive |
| Trust | Modest | | | Deep |
| Internal change | Minimal | | | Great |
| Managerial complexity | Simple | | | Complex |
| Strategic value | Minor | | | Major |
| Co-creation of value | Sole | | | Conjoined |
| Synergistic value | Occasional | | | Predominant |
| Innovation | Seldom | | | Frequent |
| External system change | Rare | | | Common |

**Figure 1.** Variables used to characterize the evolutionary nature of partnerships in the collaborative value creation framework [15].

The "partnership broker" [28] function reflects interactions across multiple boundaries and seeks to transform uncertain conditions into collaboration opportunities [10]. This function may be undertaken by an organization or by an individual (sometimes both) [28,29]. "Several studies call for an orchestrator of partnerships[ . . . ]. Most studies view orchestration as simply initiating and supporting individual partnerships. Proposed orchestrators include international institutions, government departments [13,30,31] or professional orchestrators [32]" [4] (p. 4). Key functions of the facilitating role includes generation of a collaboration context; fostering co-creation; mediation and promotion of key transversal processes such as innovation, learning, gaining wider influence, etc. [33]. In essence, this involves creating trust capital among partners [34].

The importance of "deep or radical" collaborative arrangements to ensure the transformational agenda of the SDGs [1,2] demands that we pay more attention to understanding the processes and barriers relating to partnership formation, evolution and facilitation. The following sections explore the case of EDD, a partnership that, in its first months, influenced public policies in the de-escalation and recovery of the COVID-19 crisis in Spain, through the deep interaction among a number of different organizations working in partnership through a distributed leadership model.

## 3. Research Approach

### 3.1. Research Aims and Scope

The present study focuses on the early development of the "El día después" (EDD), a multi-stakeholder partnership that includes public, private and academic parties. It seeks to deepen our understanding of how partnerships contribute to addressing systemic change. In this case, EDD represents a partnership whose purpose was to create an infrastructure within which different stakeholders involved in the response to COVID-19 could come together to address the crisis. Attention is given to how the partnership was formed and rapidly progressed to effective and impactful working at scale and the ingredients that yielded its transformational capacity. Drawing lessons from the lived experience of practice, with some of the authors being members of the partnership, enabled us to extract the critical factors underpinning effectiveness and impact. A key outcome was how the EDD

partnership moved through the lifecycle perspective, covering the early phases in a matter of weeks rather than months or even years. The EDD partnership became transformative, directly omitting the preliminary stages [15]. Here, we examine the formation and working model of EDD through the lens of the aforementioned frameworks: the partnership lifecycle phases model [24], and the collaborative value creation model [15]. This work has both theoretical and practical implications. The combination of grounded self-assessment allowed us to delve into the conceptualization of transformational and SDG-oriented partnerships. From a practical point of view, we provide recommendations to accelerate the formation, effective working and outcomes of a partnership. A key focus of our work was the facilitation of partnership working given that this role was undertaken by each of the partners as opposed to an single organization or individual [28].

### 3.2. Methodology

This investigation uses a case study methodology which is typically adopted to investigate a contemporary phenomenon ("the case") in depth and within its real-world context. Case studies offer rich empirical descriptions of specific instances of a phenomenon based on a variety of data sources [35] because they enable insights into complex cause–effect relationships that can provide useful pointers for addressing major substantive themes in a field [36]; this methodology is also useful for theory building [37]. A wide range of fields have used case studies, particularly education [38,39], management, supply chain and operations research [40–44] and, most importantly for this work, sustainability [45,46]. Case studies are singularly appropriate for analyzing collaborative initiatives because of their multi-disciplinary and cross-cutting nature [47].

We analyzed the formation of EDD through the lifecycle framework [24]; the key attributes are summarized in Table 1. Regarding CVC analysis the different variables proposed by Austin and Seitanidi have been grouped according to four categories [33], namely organizational engagement; resources and activities; partnership dynamics, and impact (see Table 2).

**Table 2.** Categories used for analysis of collaborative value creation.

| Categories | Original CVC Framework Variables |
|---|---|
| Organizational engagement | Level of engagement<br>Importance to mission |
| Resources and activities | Type of resources<br>Magnitude of resources<br>Scope of activities<br>Managerial complexity |
| Partnership dynamics | Interaction<br>Trust<br>Internal change |
| Impact | Co-creation of value<br>Synergistic value<br>Strategic value<br>Innovation<br>External system change |

The case study methodology often uses the triangulation of a set of sources of evidence to substantiate findings robustly[35]. In addition, the combination of perspectives from multiple researchers may amplify the creative potential in a case study [37]. This investigation has been conducted by six researchers, four of whom participating in EDD working teams and the other two acting as external observers. The case study was conducted from March to July 2020 and used the following sources: key documentation related to several partnership activities (including project proposals, terms of references, working documents, etc.); direct observation in the field (including attendance at EDD team meetings, EDD Communities meetings, virtual workshops and seminars,

etc.) and open interviews with selected stakeholders. The EDD partners' representatives revised and validated the final version of this paper.

## 4. The Case of "El día después"

Created in March 2020, the EDD partnership was forged to frame an SDG-oriented response to the COVID-19 crisis. EDD was formed by four different organizations, namely Iberdrola, a global company in renewable energy; itdUPM, a public university innovation center; ISGlobal, a global health research center, and SDSN Spain, the Spanish node of the Sustainable Development Solutions Network. These four organizations had collaborated before in bilateral and multi-stakeholder projects but had not worked in this particular configuration, and starting a partnership had not been on any of their agendas. The pandemic sparked a call to action for a deeper collaboration among their executives, based on mutual trust and the common willingness to innovate in collaborative arrangements. This was materialized in the multi-stakeholder partnership that became known as EDD.

The partnership started on 17 March 2020, with a first meeting among the four organizations. This first stage was characterized by strong interactions, focused on articulating the objective of the partnership—this being the opportunity to offer a collaborative response to the COVID-19 crises based on the SDGs. This represents the partnership's value proposition for EDD. The first public event presented the initiative on 25 March, when a call for collaboration of organizations to create an EDD Network was released, with the first meeting of the network held on 2 April. Just four days later, on 6 April, four communities had been created: Global Governance and Cooperation for Development; Cities; Health & Environment; Inequalities and New Economic Models. Each community comprised a core team of 10 people drawn from public and private decision-makers with civil society leaders. Communities were coordinated through one cross-community meeting and one community-specific meeting per week. In addition, three virtual spaces were opened to help communities progress and amplify their impact: (i) Agora, a hybrid space for conversations and interpretation among actors with diverse sensibilities; (ii) Workshops, a co-design space for positioning on a topic with experts from a community, and (iii) CoLab, a mass interaction space for activation of collective intelligence through prototypes. Through virtual spaces, each community held its first public event to set the vision of the community and position the initiative in a specific response to the COVID-19 area of work. On 25 May, once the communities had consolidated their direction of work and grown in number to approximately 20 members per community, which incidentally coincided with the start of the COVID-19 de-escalation and reopening process in Spain, the partnership progressed to the second stage of "maturity". A transition process was then held to reorganize the internal team and better support communities in a mid-term scenario with two priorities: to consolidate trust and shared purpose within communities and to launch the first transformative actions. Since then, communities have focused on designing demonstration projects at scale, involving stakeholders and promoting cross-learning among projects. Figure 2 synthesizes the timeline of the different EDD steps.

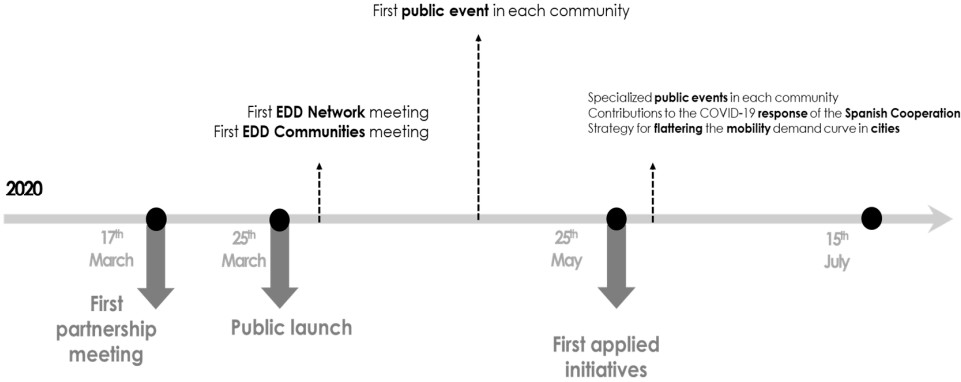

**Figure 2.** Timeline of "El día después".

*Objectives and Preliminary Results*

EDD Partnership has mobilized and consolidated a broad ecosystem of people and organizations in a very short space of time, convened around a shared purpose of framing an SDG-oriented response to the COVID-19 crisis in Spain. More than 80 experts and decision-makers with very diverse political sensitivities are attending EDD Community meetings weekly and 50 public and private organizations are now part of the wider EDD Network. In addition, public events have caught the attention of citizens, many of whom are also now participating. Thus far, more than 35,000 views have been reached at 13 public events and 150 experts have participated in three closed workshops. Outputs from the communities have served to introduce the possibilities that the pursuit of the SDGs are relevant as a means to manage the COVID-19 recovery in Spain using a multi-stakeholder perspective. For example, outcomes to date include the development of policy papers on the case for universal basic income, contributions to the Joint Response of the COVID-19 Crisis of the Spanish Cooperation and the creation of a strategy for flattening the transport mobility demand curve in cities. Although, in this paper, we are not intending to assess EDD's impact, details regarding illustrative early outcomes can be found in Appendix A. Many international organizations have shown interest in how EDD was framed, seeking to adapt this partnership model to their own context. Examples include national and international institutions such as Ibero-American General Secretariat (SEGIB), Uruguayan International Cooperation Agency (AUCI), the United Nations for Development Program (UNDP) and the Brazilian Institute for Development and Sustainability (IABS), among others. A summary of the EDD organizational model can be found in Figure 3.

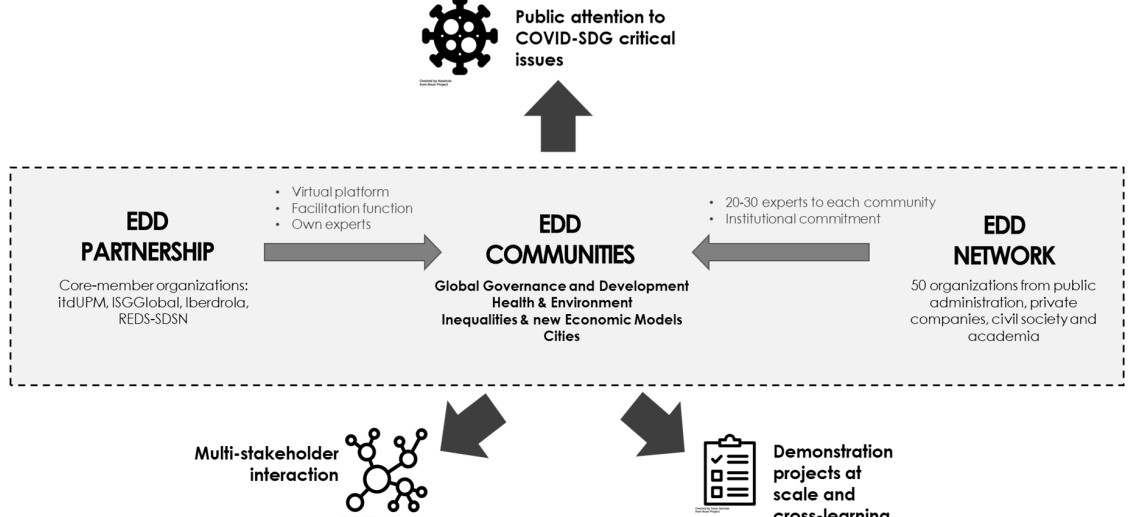

**Figure 3.** EDD organizational model.

## 5. Results: Lifecycle and Collaborative Value Creation Analysis of the EDD

This section presents the analysis of the EDD in relation to the lifecycle [24] and CVC frameworks [15], describing the differential elements that have catalyzed the EDD partnership and its maturation to transformational impact.

*5.1. Lifecycle Analysis of the EDD*

5.1.1. Scoping

The preliminary steps of a partnership are usually focused on defining its purpose. Typically, this type of collaborative arrangement seeks to tackle complex problems and define concrete challenges, breaking them down to achieve discrete and measurable results [17,21].

EDD's purpose of "offering an infrastructure within which different stakeholders involved in the response to COVID-19 can deploy their potential to collaborate" can be framed as a complex problem. However, a preliminary outcome was not defined. EDD was structured through four different but complementary communities. Its driving groups, made up of around 20 renowned professionals from academia, private companies, public administration and civil society, identified issues of shared interest and tangible opportunities for action from a multi-stakeholder and evolutionary perspective. For example, the community of "Cities" identified sustainable mobility as one of its main themes and, within it, promoted various initiatives to flatten the transport demand curve in the reopening of several Spanish cities (Madrid, Barcelona, Seville, Valencia, Santa Cruz de Tenerife, Las Palmas de Gran Canaria and Palma de Mallorca).

EDD's main objective was to accelerate systemic changes by bringing policymakers closer to scientists, industry and civil society. To do this, instead of following a linear logic supported by project planning aimed at achieving pre-established results, EDD used an evolutionary logic with a series of fundamental elements explored here: connecting strategic initiatives and people, sharing learnings incorporating "problem owners" from the start and having ambition for scale.

Once the main pupose of a partnership has been set, the potential partners usually evaluate whether an appropiate combination of their complementary assets (competencies, skills and resources) can address the identified challenge in an original manner [11,23,47,48]. Generally, a facilitation function is assigned to a designated organization or an individual to translate ambiguous conditions into collaborative opportunities [10,28,29].

EDD started with a complementary combination of organizations with some previous experience of working together, but not in the arrangement described here. Bringing in some trustful relationships among some of its members, together with some common experiences of working in collaborative environments, were foundational assets. All had exposure to the SDGs in some capacity; for example, ISGlobal is a research institute focused on global health, Iberdrola is a multinational energy company that is world leading in renewable production, and itdUPM is the innovation center at a technological university focused on partnerships for the SDGs, with a network of Spanish professionals from mainly academia and public administration commited to the 2030 Agenda.

The collaborative environment of the EDD makes the facilitation function essential. However, this function was not undertaken by a single unit, person or organization but was held by all EDD partners. Each partner contributed its added value in facilitation; some had more direct contact with a certain sector (such as private companies or science) or more developed communication skills; others had specialized knowledge in organizational innovations and multi-stakeholder work. However, all of the partners took on the role of facilitators and this distributed facilitation function has allowed EDD to create and sustain a distributed leadership culture [21,49,50]. This was deemed to be an essential element of EDD in enabling it to consolidate relationships among a wide number of diverse stakeholders with all due speed.

### 5.1.2. Initiating

When a partnership begins its activities, it is common to draw up an agreement among its members that includes the differential contributions, the governance and accountability mechanisms and the joint working structures. In this process, there is usually a tension between flexibility and formalization, and the resulting agreements are typically explicit [11,20,21,44]. A consideration of whether to increase the number of organizations that form or are related to a partnership, how best to manage the tension between maintaining control and increasing diversity, results in most cases in a formal process that can represent a drag factor on creativity and innovation. This tension will normally be present throughout all the partnership activities, but what happens during the formation stage may condition the control–flexibility balance during the lifecycle [11,20,21].

At EDD, there has been no trend towards formalization. In the five months of intense work among the partners, it has not been necessary to regulate their relationships or contributions with

any formal agreement. From the point of view of working structures, a common culture has been consolidated, but no committees or permanent structures have been formalized; people from different organizations have been mixed into diverse work teams that have evolved according to the needs of the partnership. A special emphasis has been placed on creating a shared vision among partners' representatives and across the entire EDD team. This has been developed by agile, dynamic and frequent debriefing meetings with people from all the teams involved (daily in the first two months) and through collaborative workshops when it has been necessary to define priorities or make strategic decisions. EDD needed to build relationships with a broad set of organizations to achieve impacts at scale. To facilitate their incorporation, the EDD Network was formed, which more than 50 organizations from academia, the private sector, public administration and civil society have joined through a letter of commitment (a "soft" agreement in which their alignment with the EDD vision was made).

### 5.1.3. Implementing

Some argue that the first operating actions of a partnership should be conceived as a continuous design process in which experimentation and learning allow the initial objectives to be refined [51]. Others complement this approach, pointing out the importance of a scaling up process (usually starting with a pilot project or a set of pilot actions) to remove forces that hinder collaboration [52] and to test new practices to strengthen partners' relationships and common decision-making procesess to achieve meaningful results [16].

These observations are partially reflected in the early stages of EDD's formation. Regarding the scaling-up process, EDD was not envisaged as a "quick win" pilot and opted to start its activities with demonstrator projects at scale. For example, to keep the use of public transport constant but safe in the reopening of cities, the community of "Cities" promoted an agreement to flatten the mobility demand curve in the city of Madrid, involving the main stakeholders of Madrid City Council and the Regional Government, universities, business associations and green growth companies and the main trade unions. The city released a letter that encouraged organizations to adopt a number of commitments including cutting the number of movements by their workforce by at least 30 per cent over pre-lockdown levels; introducing flexible working hours and promoting the use of public transport and cycling among their employees. The EDD team is now supporting this initiative to create a virtual platform to assist flattening the mobility demand curve and is considering expansion to other cities involved in the EDD community.

It has been essential to integrate what was already emerging in the context of the communities instead of generating actions from within the partnership's members. The active participation of the private sector and policymakers combined with academia and civil society has been crucial to frame the various transformative initiatives and to improve their chances of success. EDD promoted a culture focused on building interpersonal and organizational trust and cross-learning among the different communities.

### 5.1.4. Assessing

A key finding of the global assessment of EDD in relation to the lifecycle approach was that it has only partially followed the stages described in the theory [24]. EDD fits with those patterns described by previous literature regarding the complexity of the challenge it sought to address [17,21], the complementary combination of partners' assets [11,23,47,48], the need of a facilitation function [10,28,29], the conception of its activities as a continuous learning process [51] and the importance of action-oriented processes as a way to overcome practical barriers [53]. However, EDD has several singularities, particularly in regard to its evolutionary perspective on goal setting, the facilitation role being distributed among all the partners, the flexibility and trust-based approach to framing governance and the approach of initiating with scale projects instead of prototypes. These differential elements, summarized in Table 3, point to some key insights into how to catalyze transformative partnerships to increase effectiveness and impact; these are analyzed in the next section.

**Table 3.** EDD differential elements in relation to lifecycle assessment.

| Stages | EDD Differential Elements |
| --- | --- |
| Scoping | Evolutionary logic and distributed facilitation function among all partners. |
| Initiating | No trend towards formalization: governance or contributions based on trust, common culture but non-permanent working structures, flexible and agile new organizations' interaction. |
| Implementing | Starting with demonstration projects at scale: integrating what was already emerging in the context; active participation of private sector and policymakers, combined with academia and civil society; culture focused on building interpersonal and organizational trust and cross-learning. |

*5.2. Collaborative Value Creation Analysis of the EDD*

Throughout this section, the initial "position" of EDD will be described with respect to the key pointers defined in the CVC framework [15], grouped into four categories, namely organizational engagement; resources and activities; partnership dynamics and impact.

5.2.1. Organizational Engagement

The EDD partnership was forged in a crisis, reflecting the mission of its promoters. The 2030 Agenda has since become a central element for many organizations and is fundamental for EDD's partners. Iberdrola has incorporated the SDGs into its business strategy and corporate governance system [54]. The university vehicle of itdUPM has contributed to the development of a new SDG-aligned research strategy for Universidad Politécnica de Madrid (UPM) [55] and its governing council's commitment to decarbonizing the campus by 2030 [56]. Multi-stakeholder partnerships such as EDD are a way which itdUPM employs to reach these commitments. For ISGlobal, the SDGs are a core element of its mission, "contributing through education, research and advocacy to the challenges of global health" [57]. In addition, SDSN-Spain's mission is centered directly around the SDGs to "mobilize and sensitize Spanish society, public institutions and private [ ... ] around the SDGs, as well as favor their incorporation into public policies, the business environment and behavior of society in general" [58]. As such, the founding partners had the SDGs as a shared narrative which supported effective communication among them.

The level of organizational engagement of EDD partners was very high from the start. The managers of staff involved and their teams were aware from the first days of the COVID-19 confinement in Spain that it could have a dramatic effect on society and understood the necessity to react quickly. This created a shared purpose among partners together with a sense of urgency. The main objective for EDD was to influence the de-escalation and recovery process, leaning on the pillars of the 2030 Agenda of collaboration and transformation. All partners had strong prior experience in partnerships and other collaborative arrangements. However, EDD emerged as a unique partnership, connected with the social priorities of the moment as well as the longer-term SDGs, with a strong capacity for and interest in attracting other organizations. This has contributed significantly to maintaining a very high level of commitment for partners compared to the previous experiences of EDD members in other collaborative ventures.

5.2.2. Resources and Activities

The dedication of resources to EDD by its founders was significant from the beginning. Given the ambition of the initiative, from the first day, a team of more than 30 people from all the partners came together. EDD was a priority action for the individuals concerned and the organizations they represented. Once communities were established and consolidated, 80 experts from industry, academia, public administration and civil society joined EDD. This ecosystem of people and organizations was complex, but EDD opted for agile, evolutionary management, with no intent to formalize governance.

This was supported by flexible and frequent meetings, workshops for joint decision-making and fostering a shared work culture around the values of commitment, agility, flexibility, attention to incentives and details, generosity and distributed leadership. As aforementioned, EDD has a broad scope for its activities, centered around public policy contributions, demonstration projects at scale and mass public broadcasting activities. To carry out these activities, EDD partners combined their core competencies in facilitation, providing interdisciplinary strengths to the EDD team and to the communities' work, connecting partners' specialists to each community.

### 5.2.3. Partnership Dynamics

The level of interaction among EDD partners has been intensive from the very first days. There was a daily coordination and debriefing meeting of EDD teams (three times a week from the third EDD month), a weekly coordination meeting across all communities and another in each community, with 10 public events organized in the first two months. This level of interaction could not have been maintained without the strong organizational purpose and, above all, without high levels of trust among the organizations and the people who are part of EDD. In a simplified way, trust can be expressed as the sum of credibility, reliability and intimacy—divided by self-orientation or ego [34]. The prestige of the partner organizations reinforced the credibility of their teams and the previous relationships among them, the reliability and intimacy. Moreover, the shared purpose that was created around the EDD has led to the emergence of a strong shared identity with EDD, regardless of the partner to which each person is affiliated. As a result, deep trustful relationships among its EDD partners are a key asset of the partnership.

### 5.2.4. Impact

Co-creation and the aspiration of generating value through a collaborative process have always been essential objectives of the EDD. Delving into the generation of synergistic and strategic value, EDD sustains an innovation process highly valued by the different stakeholders. For the public administrations, the possibility of interacting in a diverse ecosystem under conditions of equivalence with all the stakeholders, and the dynamism and generosity favored by the EDD team, allowed them to advance faster and more boldly than in classic innovation networks or spaces. For private companies, the commitment of other actors to transformation represented an opportunity for more effective innovation processes was linked to the real needs of society. For the academic sector, EDD represented an opportunity for sustained dialogue with other stakeholders and the possibility of participating in multi-stakeholder projects at scale to advance knowledge sharing. Thus, having the problem owners at the center of the design (usually the public sector), and the sustained co-creation with the rest of the stakeholders, significantly increased the possibilities of systemic change in the EDD initiatives.

### 5.2.5. Assessment of the CVC Analysis of the EDD

An essential finding of the global assessment of the EDD in relation to the CVC framework was that the beginning of its activity could be characterized as transformational without the need to have gone through the previous stages. Table 4 summarizes the analysis, highlighting those EDD characteristics that have contributed differentially to the transformational nature of this partnership.

**Table 4.** Analysis of EDD using the CVC framework.

| Nature of Relationship (CVC Framework) | | Status at the Beginning of the EDD | EDD Transformational Characteristics |
|---|---|---|---|
| Organizational engagement | Level of engagement | High | 2030 Agenda as a central element for the mission of many organizations; COVID-19 urgency of reacting; EDD as a referential space. |
| | Importance to mission | Central | |
| Resources and activities | Type of resources | Core competences | Agile management based on shared values: commitment, agility, flexibility, attention to incentives and details, generosity and distributed leadership. |
| | Magnitude of resources | Big | |
| | Scope of activities Managerial complexity | Broad Complex | |
| Partnership dynamics | Interaction level | Intensive | Previous interpersonal relationships among partners' teams and shared purpose that lead to shared identity. |
| | Trust | Deep | |
| | Internal change | Medium | |
| Impact | Co-creation of value | High | Problem owners at the center of a sustained co-creation process. |
| | Synergistic value | Predominant | |
| | Strategic value | Major | |
| | Innovation | Frequent | |
| | External system change | Common | |

## 6. Discussion

From a theoretical point of view, this study shows the relevance of the combined use of well-grounded frameworks to the assessment of a partnership and the practical utility of them. The lifecycle approach [24] provides a series of valuable pointers to analyze the stages and steps necessary in the formation of a partnership. The CVC framework [15] gives a complementary vision of design elements and partner relationships to increase the transformation potential of a partnership. The comparison of the EDD analysis with both frameworks validates their fundamental approaches but also highlights some important particularities, offering insights into how to increase the efficiency and impact of working in partnerships.

Catalyzing is not only a question of speed. Critical steps that must be followed include problem framing, incentives assessment, facilitation function, joint working mechanisms, practical actions to encourage collaboration, etc. However, according to the EDD experience to date, effectiveness and efficacy can be strongly driven by following a non-linear logic, flexibility and adopting a portfolio approach, integrating what is emerging in context.

Flexibility regarding formal agreements or evolutionary management may provide agility to a partnership and a wider space for innovation but also carries certain risks, such as lack of coordination and/or misalignment [11]. At EDD, this has been compensated for by a set of actions seeking to promote a strong shared identity and directionality. Distributed leadership can also benefit from the lack of a formal agreement, since this normally also establishes a certain hierarchy. In the case of EDD, leadership has passed from one person to another dependent upon the requirement in the moment. Navigating in such conditions may be challenging for those who are not used to collaborative work with other organizations. The fact that all the partners had deep (positive) experiences in previous collaborative arrangements encouraged atypical partnership decisions, such as that of all partners choosing to act as

facilitators. Thus, the possibilities of working in an organizational context outside the partnership greatly increased, enabling EDD partners to share stakeholders and assets and connect their networks. Designing from the shared priorities of the partners and their stakeholders may reinforce effectiveness as the efforts can be concentrated on what has the most potential for systemic change. In addition, in the case of undertaking new actions, they considered what was already in progress and, based on previous learning, could respond to the interests of a wide group of organizations or orientations. However, this also poses two challenges. At first, the possible reluctance of the stakeholders to join something that has not yet shown results and, how to maintain their engagement in the medium term. To overcome the first barrier at EDD, the following attributes were considered fundamental:

- The prestige and experience of the partnering organizations.
- The digital component as an amplifier of incentives, which has allowed relevant stakeholders to be connected easily, reaching a wider audience and systematizing work in an open way.
- Identifying windows of opportunity to connect with highly relevant topics related to the emergent COVID-19 crisis and its forecast downstream impacts.

For the second challenge, the EDD support and the facilitation function were essential, allowing it to respond in an agile way to the demands posed at the EDD Communities, creating a dynamic environment, balancing reflection and action. A solid, innovative and interdisciplinary facilitating team has been essential for this mission.

From the point of view of impact and the generation of systemic change, an adequate portfolio of partnerships and the design of a common roadmap are the two main challenges highlighted around the "identification problem" [4], a missing intermediary space between bottom-up and top-down partnership approaches, needed to achieve transformation. EDD seeks to fulfill this space by reinforcing partnering capacity to its relevant stakeholders comprising policymakers, industry, civil society and academia. Making a comparison with start-up incubators, we define EDD as a partnership incubator offering its stakeholders a value proposition that can be summarized as:

- Multi-stakeholder networking, connecting spaces where relevant stakeholders interact in a context of trust and symmetry.
- Cross-learning among a myriad of people and ongoing initiatives.
- Strategic communication and advocacy, including the ability to introduce critical issues into public debate.

Through this partnership incubator approach, it is possible to create directionality and shared purpose in the work of a wide ecosystem of relevant stakeholders and a connected set of partnerships, where public policies and social priorities are at the center of the design and implementation processes.

Creating an ambitious multi-stakeholder partnership is complex and consumes resources in the forming stage. However, we assert that trustful collaborative working is necessary to address the collective systemic challenges posed by the 2030 Agenda. Based on our experiences to date, some lessons learned, or recommendations for future working, include the following:

- At the level of the individual, curiosity, humility and generosity are required behavioral attributes for people to engage in the co-creation of a shared vision and a common work culture with others; this allowed all the people involved in EDD to enjoy great autonomy and, at the same time, a strong sense of belonging and shared purpose.
- At the level of the team, the work of facilitation is essential but, as we have demonstrated, it can be undertaken by all of the partners. The role of facilitators has been decisive in creating a mutually respectful and reinforcing interdisciplinary team, with team members possessing a double organizational identity, namely identifying with both their host organization and with EDD. This fact, together with the aforementioned shared purpose, allowed knowledge transfer among the partners in terms of collaborative practices and approaches, framed as organizational innovations through EDD that could be adopted by each partner organization where appropriate.

- At the level of the community, to effect actions from the collective endeavors, it was important that committed policymakers were included in the collaboration vehicle—in this case, EDD. These translators working in innovative, dynamic, flexible and diverse multi-stakeholder co-creation processes can support the scaling of innovation to effect changes in public policy. Policymakers may also act as the commissioner and/or problem owner to help a partnership become established. In our case, EDD supported and accelerated existing or emerging policies and inspired new ones, fostering innovation in the policy making process.

An explicit limitation of this research was the fact that the conclusions are derived from a single case study. EDD provides rich and ongoing insights into how to reinforce the partnering capacity of a wide set of stakeholders, drawn together through the COVID-19 crisis and committed to accelerating the achievement of the SDGs. Regarding the theoretical frameworks used in this study, the lifecycle and CVC approach both have multiple nuances that may merit further attention. Relationships between trust building among partners and the creation of synergistic value or a portfolio approach versus pilot-scaling strategies in partnership implementation are examples of possible future areas of research.

This study shows how a crisis mobilized a commitment to the SDGs among organizations to come together in a multi-stakeholder partnership focused on long-term systemic transformation as COVID-19 served to reveal the fissures and inequities in our world. Crisis aside, these dynamics will be essential for SDG 17 to unleash its full potential to enable fulfilment of the 2030 Agenda. While there is a consensus about the importance of partnerships to deliver against shared goals, there are few practical cases of multi-stakeholder partnerships aimed at reinforcing the partnering capacity of a wide range of relevant stakeholders. For this reason, gaining insights from the practice of partnerships is essential to advance their efficiency and capacity to achieve systemic impacts.

**Author Contributions:** Conceptualization, J.M.-S.; methodology, J.M.-S., T.S.-C.; validation, W.M.P., T.S.-C., J.L.; formal analysis, J.M.-S., M.S.; investigation, J.M.-S., M.S.; writing—original draft preparation, J.M.-S., W.M.P., M.S.; writing—review and editing, J.M.-S., W.M.P., J.L., C.M.; visualization, J.M.-S.; supervision, W.M.P., T.S.-C., J.L., C.M.; project administration, J.M.-S., M.S., C.M., J.L.; funding acquisition, C.M., J.L., J.M.-S. All authors have read and agreed to the published version of the manuscript.

**Funding:** This doctoral research has been supported by funding from Universidad Politécnica de Madrid (UPM) and Iberdrola-UPM Chair on Sustainable Development Goals.

**Acknowledgments:** The authors would like to acknowledge the hard work of the people comprising the EDD facilitating team: Irene Ezquerra, Xose Ramil, Simona Perfetti, Andrea Amaya, Valentina Oquendo, Caren Camiscia, Mónica del Moral, Cecilia López, Mari Ángeles Huerta, Manuel Almestar, Marta García, Leire Pajín, Virginia Rodríguez, Gonzalo Fanjul, Laura Hidalgo, Patricia Pascau, Javier Sancho, Mónica Oviedo, Irene Schiavon, Raquel Fernández, Alfredo Azabal. The authors would also like to acknowledge the indispensable contribution of people comprising EDD's communities and the institutional commitment of UPM, Iberdrola, ISGlobal and SDSN-Spain.

**Conflicts of Interest:** The authors declare no conflict of interest. The funders had no role in the design of the study; in the collection, analyses, or interpretation of data; in the writing of the manuscript, or in the decision to publish the results.

## Appendix A

This appendix contains additional information on some of the early results reached at EDD's communities.

**Table A1.** EDD Communities' early results.

| Element | EDD Community | Description | Link |
|---|---|---|---|
| Communities' participants testimonies | All | Statements by 9 protagonists of the EDD Communities (from the public administration, business, academia and civil society) summarizing their experience and the added value of the initiative. | Video |
| Agora: "The transformation of cities" | Cities | Conversation with 14 mayors of Spanish cities to contrast their Covid-19 recovery strategies with public administration, business, academia and civil society. | Summary and video |
| Flattening of the mobility curve | Cities | Bases of the EDD proposal to reduce the mobility demand (in English). | Article |
| Madrid's main mobility stakeholder's commitment | Cities | Letter of commitment by Madrid City Council and the Regional Government, universities, business associations, green growth companies and the main trade unions. | Summary of the public event |
| *Acuerdos de la Villa* | Cities | Agreements by all the political forces of the Madrid City Council on the post-COVID-19 recovery strategy, which includes the EDD proposal to flatten the mobility curve (measure 232). | Agreements document |
| Agora: "Rethinking global cooperation and governance against COVID-19" | Global Governance and Development | Conversation with the Spanish Foreign Minister and representatives of multilateral organizations, private companies, NGOs and academia on multilateralism and international cooperation in the fight against COVID-19. | Summary and video |
| Joint Response of the COVID-19 Crisis of the Spanish Cooperation | Global Governance and Development | Results of the multi-stakeholder workshop (with the participation of 80 leading experts from the public administration, NGOs, academia and private sector) to make contributions to the COVID-19 response strategy of the Spanish Cooperation. | Contributions document |
| Science and humanitarian action | Global Governance and Development | Virtual meeting to strengthen links between key people in the Spanish scientific and humanitarian field and lay the foundations for future multi-stakeholder initiatives. | Summary and video |
| Agora: "A new company social contract for the day after" | Inequalities and new Economic Models | Multi-stakeholder conversation to frame the need and opportunities for the Spanish private sector for a response to COVID-19 in which companies create more social value. | Summary and video |
| Minimum vital income and basic income | Inequalities and new Economic Models | Insights for the adoption of minimum vital income and universal basic income in Spain. | Article |
| Specialized seminar about "Planetary health" | Health & Environment | Discussion about how we can create the same sense of urgency and levels of coordinated action to address the climate crisis and sustainable development. | Summary report |
| Analysis and proposals to the Draft Law on Climate Change | Health & Environment | Community's suggestions to the Draft Law on Climate Change and Energy Transition in Spain. | Summary report |
| Spanish Strategy of Circular Economy | Health & Environment | Results of the multi-stakeholder workshop (with the participation of 80 leading experts from the public administration, NGOs, academia and private sector) to make contributions to the Spanish Strategy of Circular Economy. | Summary report |

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
