# Peer review of "Catalyzing Transformational Partnerships for the SDGs: Effectiveness and Impact of the Multi-Stakeholder Initiative El día después"

_sustainability, doi:10.3390/su12177189_

Round 1
Reviewer 1 Report
The genesis of this paper is the attempt to destructure and analyse the right ingredient for the creation of a partnership that could become quickly effective in delivering societal impact at scale. The title and abstract are appropriate for the content of the text. Furthermore, the article is well constructed, the case study is well described, and analysis was well performed.
I believe the importance of this paper stems from the applicability of the approach to the several multi-stakeholder partnerships forged to frame an SDG-oriented collaborative response to the COVID-19 crisis, not only in Spain but also all over Europe.
To this extent, my only suggestion to the authors is to produce an extra content in the conclusions putting yourselves in the shoes of the reader, who may not live in the special environment that originated the EDD, and try to give some scalable and transferable (at policy - team -individual levels) advises to those who want to initiate a similar platform.
I am very glad the authors wrote this paper. It is a well-written, needed, and useful description of how a crisis fostered a positive reaction for a shared commitment to the SDGs, focusing on long-term systemic transformation as COVID-19 served to reveal the fissures and inequities in our world but also the opportunities to solve them together.
Author Response
Dear reviewer,
Thank you very much for your revision and for sharing your insights and comments. We have included some paragraphs in the conclusions section to provide final recommendations at policy, team and individual level.
We really appreciate your comments regarding the pertinence of the article.
Kind regards,
Jaime Moreno-Serna.
Reviewer 2 Report
This is a fine contribution to the literature and it is nicely presented. There are a few paragraphs in section 6 that need to be indented. Otherwise, the paper is ready for publication.
I thought the paper was fine as it was and so there was no additional feedback to provide.
Author Response
Dear reviewer,
Thank you for your revision and for sharing your insights and comments. We have corrected the paragraphs you mentioned.
Kind regards,
Jaime Moreno-Serna.